# Parameter Study of Interfacial Capacities for FRP–Steel Bonded Joints Based on 3D FE Modeling

**DOI:** 10.3390/ma15217787

**Published:** 2022-11-04

**Authors:** Jie Liu, Yu Yuan, Libin Wang, Zhongxiang Liu, Jun Yang

**Affiliations:** 1College of Civil Engineering, Nanjing Forestry University, Nanjing 210037, China; 2School of Transportation, Southeast University, Nanjing 210096, China; 3School of Civil Engineering, Suzhou University of Science and Technology, Suzhou 215129, China

**Keywords:** adhesive, equivalent stress, fiber-reinforced polymer–steel, finite element model, peeling stress

## Abstract

This paper investigated the stress distribution of an adhesive layer for GFRP–steel bonded joints under 22.48 kN tensile loading using a three-dimensional numerical simulation. Firstly, a stress analysis of three paths was conducted, and after comparison, path II (through the middle layer of the bonding layer) was adopted as the analyzing path. Furthermore, a systemically parametric study of the effects of the FRP stiffness (i.e., elastic modulus and thickness), bonding length, adhesive thickness, and adhesive modulus was conducted. For the joints with different FRP elastic moduli, the minimum value of normal peeling stress was calculated as −3.80 MPa by the FRP for 10 GPa, showing a significantly severe stress concentration of FRP for 10 GPa. An analysis of the von Mises stresses proved that the increase in FRP stiffness could reduce the stress concentration of the adhesive layer effectively. The study of the effect of bonding lengths indicated that a more uniform peeling stress distribution could result from the longest bonding size; the largest peeling stress of 6.54 MPa was calculated for a bonding length of 30 mm. Further parameter analysis showed that the stress concentration of the adhesive layer could be influenced by the FRP thickness, bonding thickness, and elastic modulus of the adhesive layer.

## 1. Introduction

A large number of steel structures such as buildings, bridges, offshore platforms, and large mining equipment (wind turbines for instance) require repairing/retrofitting due to deterioration under loading and environmental effects [1,2]. The conventional repairing (or retrofitting) method for steel structures is to cut out and replace plating or attach (or weld) external steel plates. In recent years, with the successful use of artificial fiber-reinforced polymer (FRP) in the aerospace industry, FRP reinforcement and its repairing technique have attracted extensive attention in engineering steel structures due to FRP’s superior characteristics such as a high strength-to-weight ratio, easy fabrication, superior corrosion/fatigue resistance, and good durability [3,4,5,6]. Many researchers [7,8,9] have mentioned that FRP reinforcement could effectively improve/repair the bearing capacity of engineering steel structures, including the static bearing capacity, fatigue resistance, etc.

For composite structures (i.e., structures composed of different component materials), the bonding layer is often the weak connection area [10,11]; debonding of FRP from a steel substrate is one of the main failure modes [3]. Therefore, to achieve a superior capacity of reinforced steel structures, the bonding behavior between a steel matrix and FRP needs to be studied in depth.

In 2001, Miller et al. [12] studied the bonding performance of carbon-fiber-reinforced polymer (CFRP) on a steel plate and found that the force transfer length was a function of the geometric and material properties of the steel substrate, CFRP reinforcement, and adhesive. Fawzia et al. [13] studied the bonding behaviors of FRP–steel double-strap bonded joints and mentioned that the bonding capacities and failure modes could be affected by the elastic modulus and bonding length significantly. Yu et al. [14] discussed the performance of CFRP-to-steel bonded interfaces based on an experimental study and concluded that nonlinear adhesives with a lower elastic modulus but a larger strain capacity were more beneficial than linear adhesives with a similar or even higher tensile strength. Neto et al. [15] conducted a parametric study of adhesive joints with composites and discussed the failure in adhesive joints with different bonding parameters. Furthermore, in 2016, Wang et al. [16] presented an experimental study on the behavior of CFRP-to-steel bonded joints with a ductile adhesive by testing single-strap pull joints using an approximately trapezoidal shape for bonded joints. He et al. [3] carried out numerical modeling of bond behavior between steel and CFRP laminates with a ductile adhesive and proposed a model for the effective bond length for steel–CFRP bonded joints.

The above researchers studied the performance change law of FRP–steel bonded structures based on macro performance analysis methods. However, when considering the complexity and difficulty in the quantitative detection of the research on the interface characteristics of the bonding structure (such as the stress distribution inside the bonding layer, the internal trend of failure, etc.) [17,18,19,20,21], the current research on the failure characteristics and change trends of micro details of the bonding layer is still relatively limited. To fill this gap, this paper presents a study of the interfacial stress for GFRP–steel double-strap bonded joints via the finite element (FE) method. Based on this study, the peeling stress and von Mises stress of the adhesive layer under different bonding parameters are analyzed and the failure mechanism of the steel–FRP bonding interface is discussed and expounded upon rationally.

## 2. Bonding Materials

### 2.1. Material Properties of GFRP and Steel

For this study, a GFRP plate fabricated using glass fibers and unsaturated polyester resin was adopted. The steel substrates were composed of Q345b steel. Based on the tensile tests of steel coupons, an average yield strength and Young’s modulus of 291.3 Mpa and 204.0 Gpa were measured, respectively. The main mechanical properties of the GFRP, adhesive, and steel are listed in Table 1.

### 2.2. Tensile Test of the Adhesive Specimen

The methacrylate adhesive material was the super commercial two-part structural adhesive PLEXUS MA 310 ( with a fixture cure time of 55 min at 25 °C and a specified operating temperature range of 55 °C to 121 °C.), which was manufactured by ITW Performance Polymer (Wujiang) Co., Ltd. in Wujiang, China. Taking into account that the adhesive used in FRP–steel bonded joints is significantly affected by the environment, the adhesive’s properties in a field environment were worth testing specifically. A unidirectional quasistatic tensile test of the adhesive specimen was conducted in the authors’ previous study [2]. The uniform specimens of the adhesive were formed into a dog-bone shape with a thickness and testing length of 15 mm and 100 mm, respectively. According to this tensile test, the mechanical behaviors of the adhesive could be adopted as shown in Figure 1. Based on the tensile test of the adhesive specimens, we concluded that the tensile strength of the adhesive was 18.85 MPa. With the increase in the specimen length, the tensile force increased nonlinearly until it fractured. To simulate the adhesive material, a secant elastic modulus (213.24 MPa) of the adhesive material was adopted.

## 3. Tensile Property of GFRP–Steel Specimen

To investigate the bonding performance of the GFRP–steel composite structure, GFRP–steel double-strap joints were manufactured and tensile-tested using a servo-hydraulic test machine (MTS Landmark with a maximum load capacity of 50 kN, Eden Prairie, MN, USA) as shown in Figure 2 and presented in the authors’ previous study [22]. The thickness of the steel plate and GFRP for the standard specimen was 5 mm, and the thickness of the adhesive (bonding layer) was 1 mm, as Figure 2a shows. During the static tensile testing, the load was applied through the two ends of the specimen with the displacement-control mode set at a rate of 1 mm/min and a tensile capacity of 22.48 kN [22].

## 4. Finite Element Model

### 4.1. The Geometry of the Finite Elements

Although experimental research can obtain more intuitive and reliable conclusions, it is difficult to monitor the internal mechanical characteristics and change mechanisms of a component. Benefiting from its convenience and economy, the finite element analysis method makes up for the shortage in experimental research methods [10,11,23,24,25,26,27], so it is widely used in the engineering research field.

To analyze the interfacial stress distribution of the GFRP–steel joints with different bonding parameters, three-dimensional finite element models were developed by using the FE software package ANSYS 14.5, as shown in Figure 3. For all finite element analyses, an implicit calculation method was performed. While taking into account the symmetry of the specimen ¼ structural models were established. The GFRP plates, adhesive layers, and steel plates were modeled using a 3D eight-node solid element (i.e., Solid 185 in ANSYS) with three degrees of freedom (DOFs) at each node (translations in the *x*, *y,* and *z* directions) and meshed with a size of 0.5 × 0.5 × 0.5 mm. Under this meshing strategy, the bonding layer was simulated by a two-layer hexahedral element.

The steel, adhesive, and GFRPs, which were assumed to be homogeneous and linearly elastic, were defined according to the values given in Table 1. To simulate the bonding effects between different components of the specimen, glue operation was used in the models. Constraints were applied to the nodes of the left end faces for the GFRP plates, and the tensile loading was simulated by applying an external load to the right end faces of the steel plate models, as Figure 3 shows.

### 4.2. Validation of the FE Model

To validate the effectiveness of the FE model, a tensile test result for a standard GFRP–steel bonded joint cured in air conditions was adopted and compared. The tensile test of the GFRP–steel bonded joint was reported in the authors’ previous study [22]. According to the tensile test, the load–displacement curve and ultimate tensile strength (22.48 kN) were adopted. Furthermore, based on the tensile test, a typical cohesive failure mode was observed for the bonded joint, which meant that the fracture was inside the adhesive material rather than at the adhesive–structure interface [22]. Based on the results of the quasistatic tensile test, we observed that the fracture was initiated at the adhesive near the gap area of the specimen; that is to say, as the load increased, the cracking propagated toward the free end of the GFRP along the bonding layers and eventually resulted in a complete disengagement of the GFRP plate from the steel substrate. Therefore, we considered that for the proposed bonded joint, the composite specimen began to fail only when the stress of the adhesive layer reached its failure strength.

While taking into account that the tensile capacity of the GFRP–steel bonded joint depended on the adhesive layers (for the cohesive failure mode) and an ultimate tensile strength of 22.48 kN was obtained in the tensile test, the FE model of the standard joint was loaded with 22.48 kN and calculated. Figure 4 shows the stress contours of the adhesive layer; it can be seen that the numerical results agreed with the experimental results well. By applying a 22.48 kN tensile loading, the maximum stress of the adhesive layer for the FE model was 18.80 MPa, which was very close to the failure strength (18.85 MPa) of the adhesive material shown in Figure 1 and Figure 2. Therefore, we believe that the model strategy could simulate the tensile properties of the GFRP–steel double-strap bonded joints effectively.

For the double-strap bonded specimen, the force transmission route under tension was divided into two paths and transmitted to the two FRP plates through the shear stress of the adhesive layer. Considering that the shear path transmitted by the adhesive layer did not coincide with the central axis of the FRP plate, therefore, a normal stress perpendicular to the interface (called the peeling stress) appeared in the strap joint area when the bonding structure was stretched; this peeling stress was the key reason for the interlaminar failure of the bonding specimen [28,29]. To analyze the distribution of the interfacial peeling stress, the normal stress of the bonding zone was extracted using path analysis technology, as shown in Figure 5: three extraction paths were respectively defined to extract the normal (Z-direction) stress at the center interface center of the FRP–adhesive layer, the Z direction stress at the middle layer of the adhesive layer, and the normal (Z direction) stress at the center interface of the steel–adhesive layer.

Figure 6 shows the normal (Z-direction) peeling stress distribution curve extracted from the three paths of the bonding zone. Based on Figure 6, we observed that the peeling stress extracted from the three paths was very close. These maximum values were 2.77; the three minimum values stresses were −3.17, −3.10, and −3.01 MPa, respectively; therefore, to facilitate the comparative analysis, path II was used as the analysis path in the subsequent path analysis. Furthermore, when analyzing the stresses of the node for a different location, the peeling stress near the left side (the center line of the test specimen) and right side were found to be much greater than the others.

Figure 7 shows the von Mises stresses of nodes on the three paths. Compared to the peeling stress distribution curve, a different change law was found according to the von Mises stress curve; the stress value decreased with an increase in the coordinate value. The maximum stress (14.71 Mpa) of the bonding layer was calculated near the centerline. This phenomenon indicated that the stress concentration at the left end of the adhesive was much greater than that of the others, which explained the failure mechanism of the bonded specimen observed previously [22].

## 5. Parametric Studies

According to previous studies by researchers, it was found that the tensile capacities of the double-strap GFRP–steel bonded joints were influenced by the FRP stiffness, bonding length, and other adhesive bonding parameters [1,30,31]. Therefore, the proposed FRP model was applied to investigate the effects of these parameters on the double-strap GFRP–steel bonded joints. The workflow of the parametric studies is shown in Figure 8.

### 5.1. Effect of FRP Elastic Modulus

Based on the producing materials, FRP can be divided into four categories: glass-fiber-reinforced polymer (GFRP), carbon-fiber-reinforced polymer (CFRP), basalt-fiber-reinforced polymer (BFRP), and aramid-fiber-reinforced polymer (AFRP). Different material properties are shown by these composites, among which the impact of the material stiffness (including elastic modulus and thickness of the FRP) on the performance of the bonded structure has received widespread attention [32,33]. To investigate the effect of the FRP’s elastic modulus on the FRP–steel bonded joints, the FRP moduli E*_f_* = 10 Gpa, 50 Gpa, 100 Gpa, 200 Gpa, and 400 Gpa were adopted in this study; the other parameters of the model were set as shown in Table 1 and Figure 3a. The results of the numerical simulation analysis are listed in Table 2 and Figure 9 and Figure 10.

Figure 9 shows the peeling stresses of the adhesive for the FRP–steel bonded joints with different FRP elastic moduli (i.e., E*_f_* = 10 Gpa, 50 Gpa, 100 Gpa, 200 Gpa, and 400 Gpa) under a 22.48 kN tensile loading. According to Figure 9, it was found that the elastic modulus of the FRP could significantly affect the normal peeling stress distribution of the bonding layer. The normal peeling stress distribution of the bonding layer was more uniform for the specimens bonded with FRP materials with a higher elastic modulus. The minimum value of the normal peeling stress was calculated as −3.80 Mpa for the FRP with an elastic modulus of 10 Gpa; as the elastic modulus of the FRP increased, the ultimate value of the normal peeling stress increased gradually. The changes in the extreme normal peeling stress showed that the stress concentration at the end of the bonding layer decreased with an increase in the FRP plate stiffness and that the bonding strength of the joint could be improved accordingly [14]. Furthermore, for the FRP with 10 Gpa, the variation range of the normal stresses of the adhesive layer was −3.80~2.44 Mpa, whereas the variation range of the bonding condition for 500 Gpa was −1.28~2.22 Mpa.

Figure 10 plots the von Mises stresses on path II for different FRP elastic moduli under a 22.48 kN tensile loading. With the increase in FRP stiffness, the von Mises stresses on the left side of the adhesive layer decreased gradually, while the stresses on the right side increased gradually. The extreme value of the von Mises stress for the left-side was 16.96 Mpa (for 10 Gpa), which was much higher than that of the specimen for 500 Gpa, and the excess ratio was about 83%. However, for the right side, the extreme value (10.18 Mpa) of the von Mises stress was calculated for the specimen with 500 Gpa, which was much higher than that of the specimen with 10 Gpa. This phenomenon showed that the change in the elastic modulus of the FRP could improve the distribution of the equivalent stress (i.e., von Mises stress) in the bonding area. In other words, an FRP with a higher elastic modulus could make the equivalent stress more uniform, which may have effectively reduced the stress concentration in the bonding end zone.

### 5.2. Effect of FRP Thickness

This section investigates the effect of the FRP’s thickness on the tensile behaviors of the bonded structures. FRP–steel double shear specimens with different FRP thicknesses (i.e., 3 mm, 4 mm, 5 mm, 6 mm, and 7 mm) were simulated and analyzed. The elastic moduli and dimensions (except for FRP thickness) of the bonding joints were set according to Table 1 and Figure 2a.

Figure 11 presents the peeling stresses on path II for different FRP thicknesses under a 22.48 kN tensile loading. Based on Figure 11, we observed that the peeling stress could be influenced by FRP thickness. As the bonding length increased, the extreme peeling stress on the right side increased accordingly. The largest peeling stress was calculated for the FRP thickness of 7 mm, which had an extreme stress of 3.50 MPa on its right side. Furthermore, for the left side, the smallest value (−3.58 MPa) of the peeling stress was calculated for the 3 mm bonding specimen, while the largest extreme peeling stress on the left side for the 4 mm, 5 mm, 6 mm, and 7 mm specimens was −3.31, −3.10, −2.96, and −2.87 MPa, respectively, showing that the change in FRP thickness likely may not have affected the left-side stress remarkably.

Figure 12 plots the von Mises stresses of the adhesive layer for different FRP thicknesses under a 22.48 kN tensile loading. We observed that the FRP thickness may have affected the extreme equivalent stress of the adhesive layer. The extreme values of the five FRP plates were 17.40 (3 mm), 15.72 (4 mm), 14.64 (5 mm), 13.89 (6 mm), and 13.34 MPa (7 mm). Compared with the other specimens, the largest extreme stress was adopted in the model of the 3 mm FRP, which indicated that the increase in the FRP thickness would reduce the extreme equivalent stress of the bonding layer. By comparing Figure 10 and Figure 12, similar trends of stress curves could be found; for example, the stress distribution of specimens made with the more rigid FRP plates (e.g., 7 mm FRP and 500 GPa) was more uniform than that of the more flexible FRP plates (e.g., 3 mm FRP and 10 GPa).

### 5.3. Effect of Bonding Length

The bonding parameter (for example, the bonding length) is another important factor that may affect the strength of bonded structures [2]. Therefore, to study the influence of bonding length on the tensile performance of the bonded joints, FRP–steel double-strap specimens with different bonding lengths (i.e., 30 mm, 60 mm, 90 mm, 120 mm, and 150 mm) were simulated and analyzed. The elastic moduli and dimensions (except for bonding lengths) were set according to Table 1 and Figure 2a.

Figure 13 plots the peeling stresses of the adhesive layer for different bonding lengths under a 22.48 kN tensile loading. We found that the bonding length may have influenced the peeling stress of the adhesive layer significantly. The largest peeling stress was calculated for a bonding length of 30 mm, which had an extreme stress of 6.54 MPa on its right side. As the bonding length increased, the extreme peeling stress decreased accordingly. Furthermore, for the left side, the smallest value (−4.50 Mpa) of the peeling stress was calculated for the 30 mm bonding specimen, which showed that a longer bonding size would lead to a more uniform peeling stress. The homogenization of the peeling stress could effectively improve the strength of specimens; similar conclusions were reached and reported in [34]. Therefore, to reduce the extreme value of the peeling stress for the adhesive layer, the bonding length should be appropriately increased. In addition, for the specimens with a bonding length greater than 90 mm, the increase in the bonding length did not have such a remarkable effect on the bonding stresses; for example, the extreme peeling stress of the 150 mm bonding specimen (0.12 Mpa) was only 0.60 Mpa less than that of the 90 mm bonding specimen (0.72 Mpa).

Figure 14 shows the von Mises stresses of the adhesive layer for different bonding lengths under a 22.48 kN tensile loading. According to the distribution of the von Mises stresses, we observed that a larger bonding length could reduce the equivalent stress of the adhesive layer significantly. The stresses of the specimen with a 30 mm bonding length were much greater than those of the other specimens. For instance, for the specimen with a 30 mm bonding length, the extreme stresses were 19.53 Mpa and 13.90 Mpa, whereas extreme stresses of 10.63 Mpa and 0.93 Mpa were calculated for the 150 mm bonding specimen, which indicated that the increase in the bonding length would reduce the equivalent stress of the bonding layer. In addition, the extreme stresses for the 90 mm bonding specimen were calculated as 12.07 Mpa and 2.85 Mpa, and quite low stress values were found. Therefore, although a longer bonding length would lead to a more appropriate and uniform adhesive stress, a bonding length for the double-strap specimen of 90 mm is recommended when considering the manufacturing costs.

### 5.4. Effect of Bonding Thickness

To study the influence of the bonding thickness on the tensile performance, FRP–steel double-strap bonded joints with different thicknesses (0.5, 1, 1.5, 2, and 2.5 mm) of the adhesive layer were modeled and analyzed. The elastic moduli and dimensions (except for the bonding thickness) were set according to Table 1 and Figure 2a.

Figure 15 plots the peeling stresses of the adhesive layer for different bonding thicknesses under a 22.48 kN tensile loading. For different bonding thicknesses (i.e., 0.5, 1.0, 1.5, 2.0, and 2.5 mm), the minimum peeling stresses of the five models were −3.71, −3.10, −2.86, −2.77, and −2.74 MPa respectively; the maximum peeling stresses of the five models were 2.08, 2.77, 3.09, 3.30, and 3.44 Mpa, respectively. The extreme difference in peeling stress for each model was 5.79, 5.87, 5.95, 6.07, and 6.18 Mpa, respectively, indicating that the difference in bonding thickness likely may not have affected the nonuniformity of the interface peeling stress. By analyzing the stress values of the different bonding thicknesses, we found that the change in the bonding thickness could adjust the peeling stress of the interface to a certain extent. This was specifically reflected by the fact that the thinner the bonding layer was, the more prominent the extreme value of peeling stress on the left side (near the centerline) was, and conversely, the thicker the bonding layer was, the more prominent the extreme value on the right side was.

Figure 16 shows the von Mises stresses of the adhesive layer for different bonding thicknesses under a 22.48 kN tensile loading. It can be seen from the curves that for each specimen with a specific bonding thickness, with an increase in the node coordinates, the equivalent stress at the interface showed a decreasing trend, showing that under the current specimen parameters (i.e., bonding length and stiffness), the early failure of the left bonding zone was the main cause of the failure of the composite structures. Furthermore, according to the distribution of the von Mises stresses for the different bonding thicknesses, we observed that the change in the bonding thicknesses could affect the equivalent stresses remarkably. The largest value of the equivalent stress was 18.14 Mpa (in the 0.5 mm bonded model); the four extreme equivalent stresses for bonding thicknesses of 1.0, 1.5, 2.0, and 2.5 mm were 14.64, 13.26, 12.60, and 12.31 Mpa, respectively, all of which were calculated on the left side of the adhesively bonded models. We concluded that with an increase in the bonding thickness (from 0.5 mm to 2.5 mm), the extreme value of the bonding layer stress decreased gradually, indicating that the increase in the bonding layer thickness could effectively reduce the stress concentration in the bonding area and increase the strength-bearing capacity of the specimen accordingly.

### 5.5. Effect of Adhesive Stiffness

In this section, the effects of the stiffness of the adhesive material are discussed based on the bonded specimens. The model strategies and parameters (except for the elastic modulus of the adhesive layer) followed were according to Table 1 and Figure 2a. Five elastic moduli (200, 500, 800, 1100, and 1400 MPa) of the adhesive were adopted; the analyzed results of the peeling and von Mise stresses of the adhesive layers are presented in Figure 17 and Figure 18.

Figure 17 plots the peeling stresses of the adhesive layer for different adhesive moduli under a 22.48 kN tensile loading. We found that the elastic modulus of the adhesive layer had a significant influence on the peeling stress of the bonded specimen. For different adhesive moduli (i.e., 200, 500, 800, 1100, and 1400 MPa), the maximum peeling stresses of the right side were 2.79, 2.38, 2.12, 1.96, and 1.86 MPa, respectively, showing that the elastic modulus of the adhesive layer had little influence on the free end (i.e., the right side of the bonded specimens). However, for the left side, the minimum peeling stresses were −3.02 (200 MPa), −4.53 (500 MPa), −5.53 (800 MPa), −6.24 (1100 MPa), and −6.75 (1400 MPa) MPa, respectively, indicating that the peeling stress on the left side of the bonding layer was more sensitive to the stiffness of the adhesives.

Figure 18 presents the comparison of the von Mises stresses of the adhesive layers for different elastic moduli under a 22.48 kN tensile loading. Similar to the results under other conditions, the difference in the stresses on the right side of the adhesive layer was quite small, whereas a remarkable stress concentration was observed on the left side of the adhesive layers, showing that the stress concentration of the left bonding layer was the main cause of the failure of the bonding zone of the specimen. The maximum equivalent stresses of the five models were 14.39 (200 MPa), 19.22 (500 Mpa), 22.68 (800 Mpa), 25.36 (1100 Mpa), and 27.55 (1400 Mpa) Mpa, respectively, indicating that the bonding materials with a higher elastic modulus were more likely to cause stress concentration in the bonding zone. Therefore, we concluded that with the same bonding parameters and conditions, the more flexible bonding material was more beneficial to the stress distribution in the bonding zone.

## 6. Conclusions

This study investigated the effects of different bonding parameters on the bonding behaviors of FRP–steel double-strap bonded structures. During this study, the effects of FRP stiffness, bonding length, adhesive thickness, and adhesive modulus were discussed, and the following conclusions were drawn:When taking into account that the thickness of the bonding layer of the bonded specimen was relatively small, the stress results (for the interface peeling stress and the von Mises stress) of different calculating paths of the bonding material were very close. Therefore, path II (through the middle layer of the bonding layer) was used as the subsequent stress analysis path.The study of different FRP stiffnesses (elastic modulus and thickness) showed that the normal peeling stress and von Mises stress distributions in the bonding layer were more uniform in the specimens bonded with more rigid FRP materials, while the FRP with a higher stiffness was more conducive to eliminating the stress concentration in the adhesive layer.An increase in the bonding length could effectively reduce the stress concentration in the adhesive layer. When considering the manufacturing costs, we recommend a bonding length of 90 mm for double-strap bonded specimens.A difference in bonding thickness likely may not affect the nonuniformity of the interface peeling stress. However, a change in the bonding thickness can affect the equivalent stress remarkably; with an increase in the bonding thickness (from 0.5 mm to 2.5 mm), the extreme value of the bonding layer stress decreased gradually.The elastic modulus of the adhesive layer had a significant influence on the peeling stress of the bonded specimen. The peeling stress on the left side of the bonding layer was more sensitive to the stiffness of the adhesives; bonding materials with a higher elastic modulus were more likely to cause stress concentration in the bonding zone. This research included a detailed analysis of the influence of various bonding parameters on the tensile behaviors of FRP–steel double-strap bonded structures; however, when considering the relatively complex failure mechanisms of bonded composite structures, a more profound experimental analysis (including different bonding parameters and conditions) is required. It is worth noting that this study was aimed at the bonding stresses of FRP–steel double-strap bonded joints based on a linear elastic assumption and cohesive failure mode; therefore, the failure mechanisms of FRP–steel composite specimens composed of nonlinear materials with other failure modes (except for cohesive failure) require further research.

## Figures and Tables

**Figure 1 materials-15-07787-f001:**
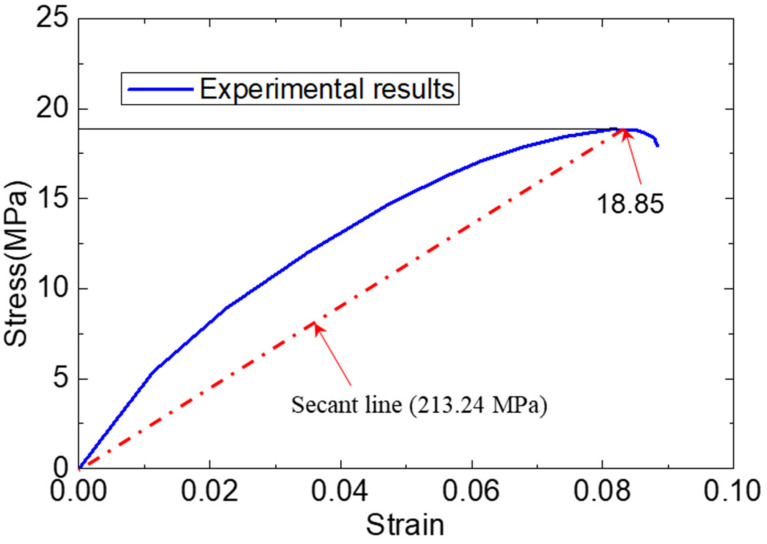
Mechanical behavior of adhesive in tensile experiment and finite element model.

**Figure 2 materials-15-07787-f002:**
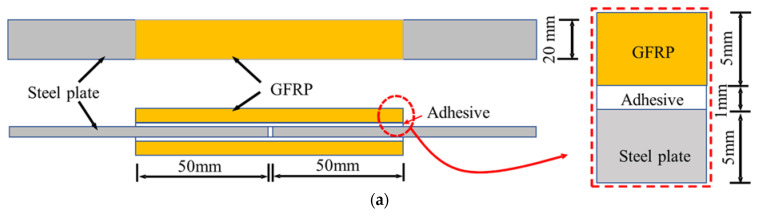
The GFRP–steel double-strap joints and the tensile test graph. (**a**) Dimensions of the GFRP–steel double-strap joints; (**b**) raw materials and test loading graph of the double-strap joint.

**Figure 3 materials-15-07787-f003:**
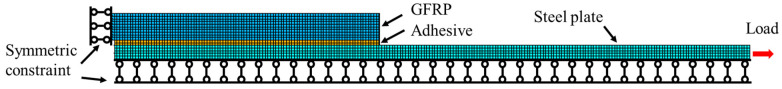
Finite element model of GFRP–steel bonded joint.

**Figure 4 materials-15-07787-f004:**
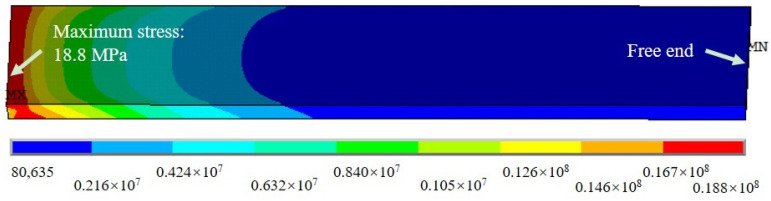
The von Mises stresses of the adhesive layer for the tensile joint.

**Figure 5 materials-15-07787-f005:**
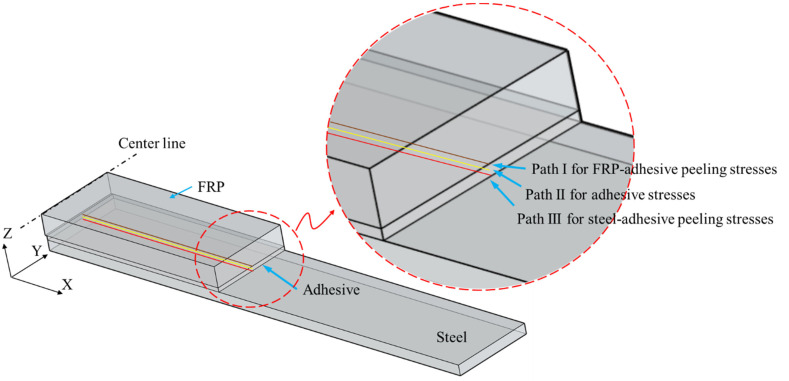
Paths for calculated adhesive stresses.

**Figure 6 materials-15-07787-f006:**
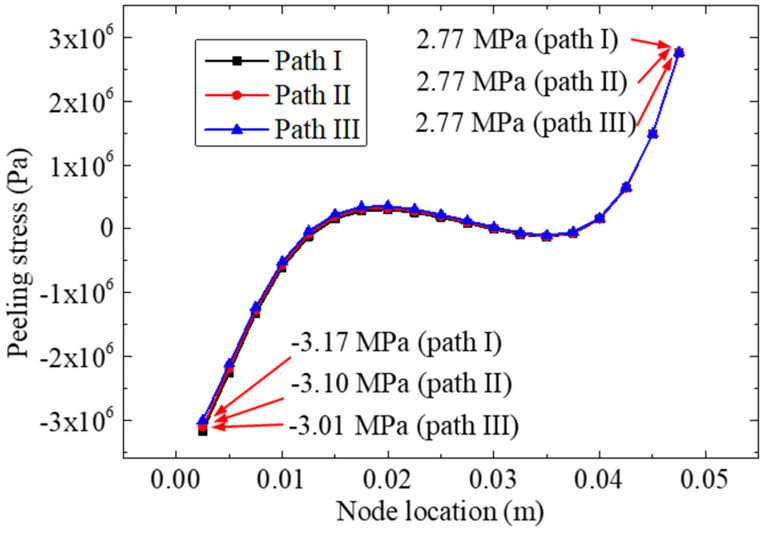
Peeling stresses distribution curve for the three paths.

**Figure 7 materials-15-07787-f007:**
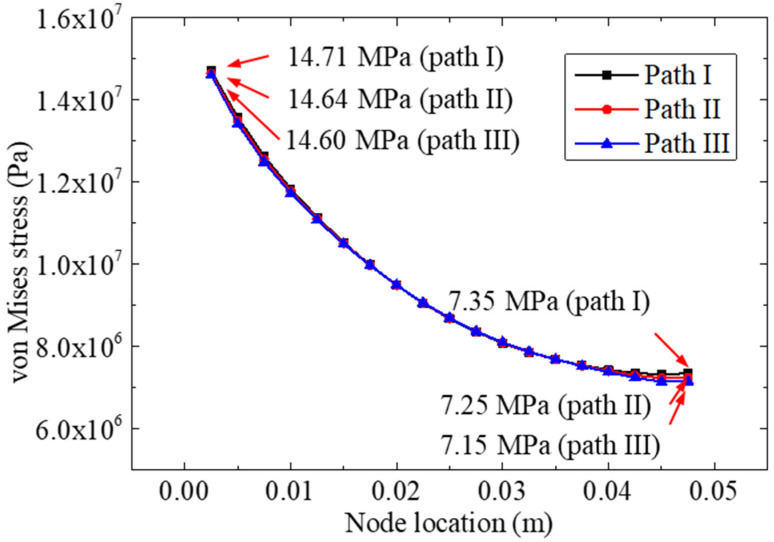
The von Mises stresses of nodes on path II.

**Figure 8 materials-15-07787-f008:**
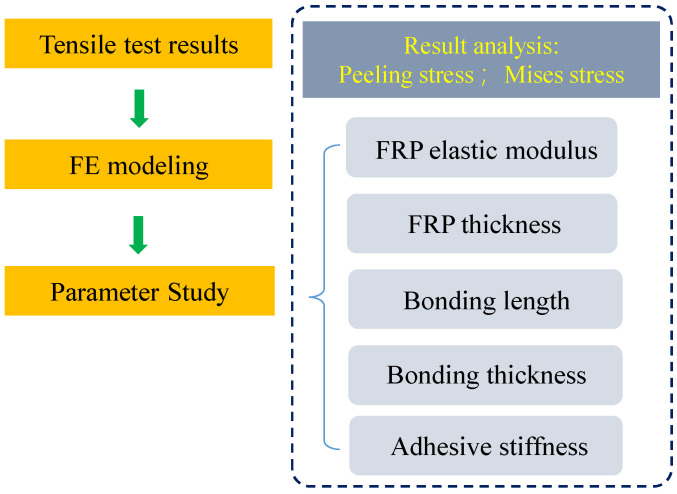
Workflow of the parametric studies.

**Figure 9 materials-15-07787-f009:**
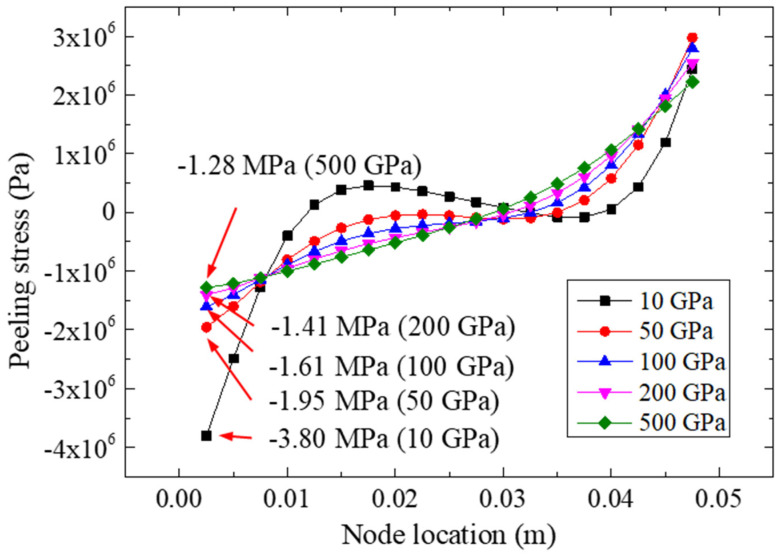
Peeling stresses on path II for different FRP elastic moduli.

**Figure 10 materials-15-07787-f010:**
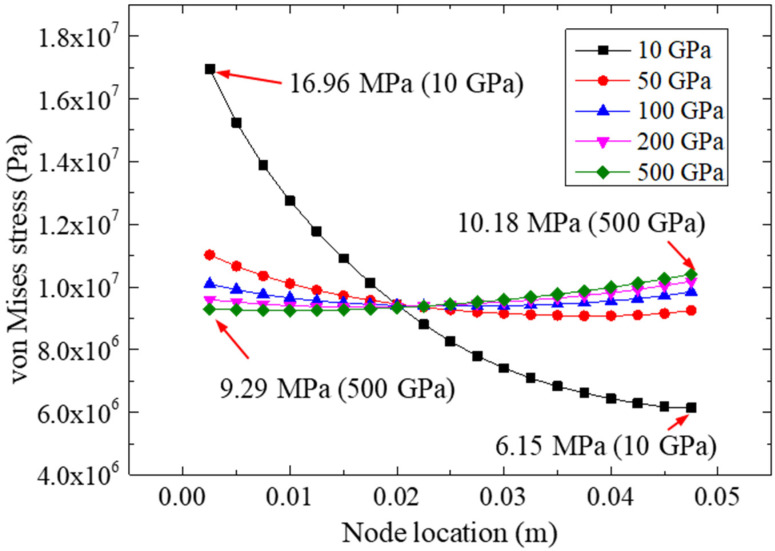
The von Mises stresses on path II for different FRP elastic modulus.

**Figure 11 materials-15-07787-f011:**
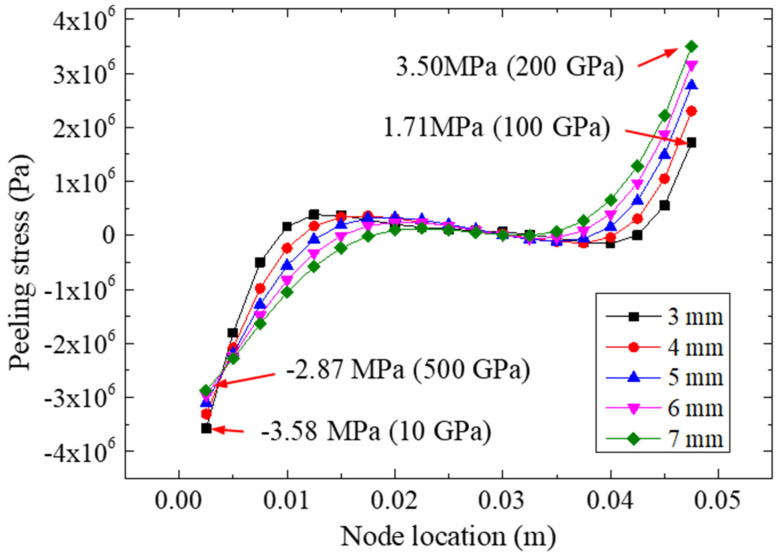
Peeling stresses on path II for different FRP thicknesses.

**Figure 12 materials-15-07787-f012:**
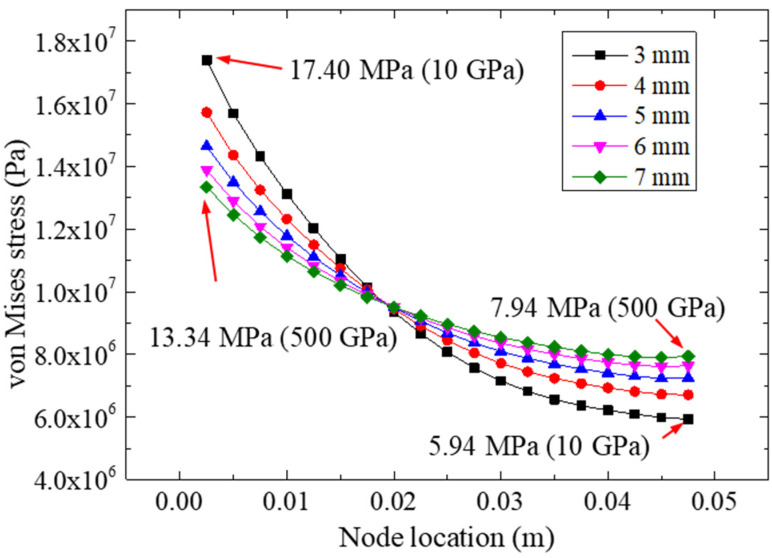
The von Mises stresses for different FRP thicknesses.

**Figure 13 materials-15-07787-f013:**
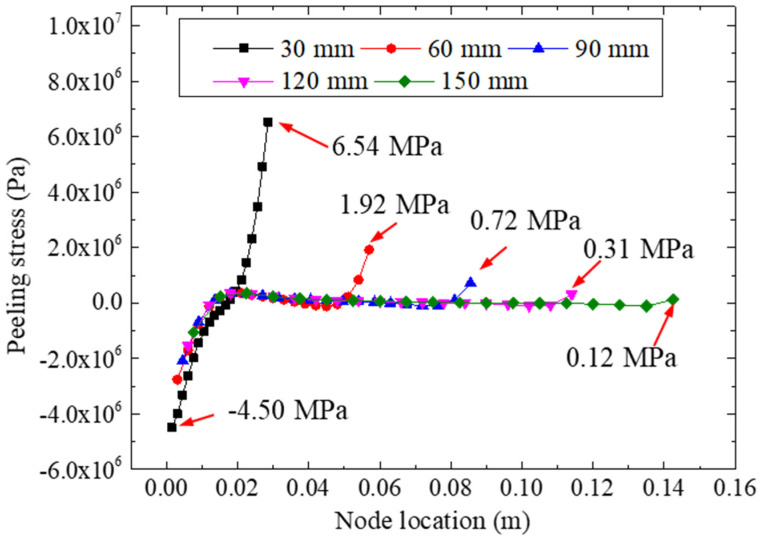
Peeling stresses for different bonding lengths.

**Figure 14 materials-15-07787-f014:**
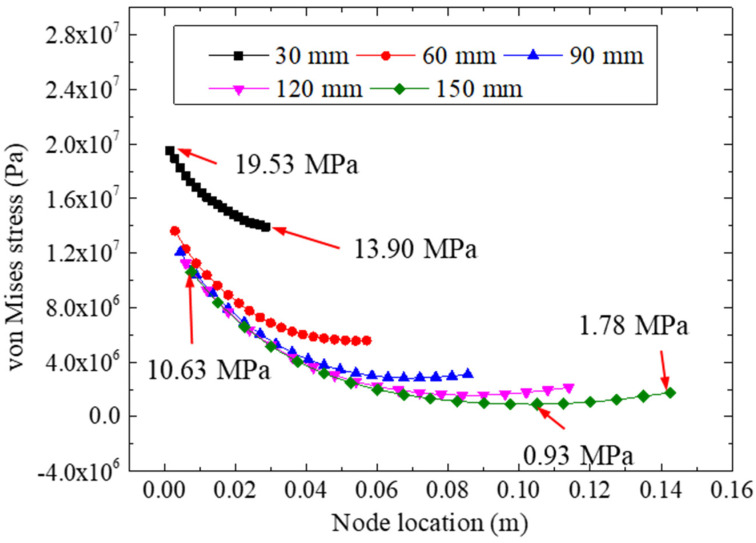
The von Mises stresses for different bonding lengths.

**Figure 15 materials-15-07787-f015:**
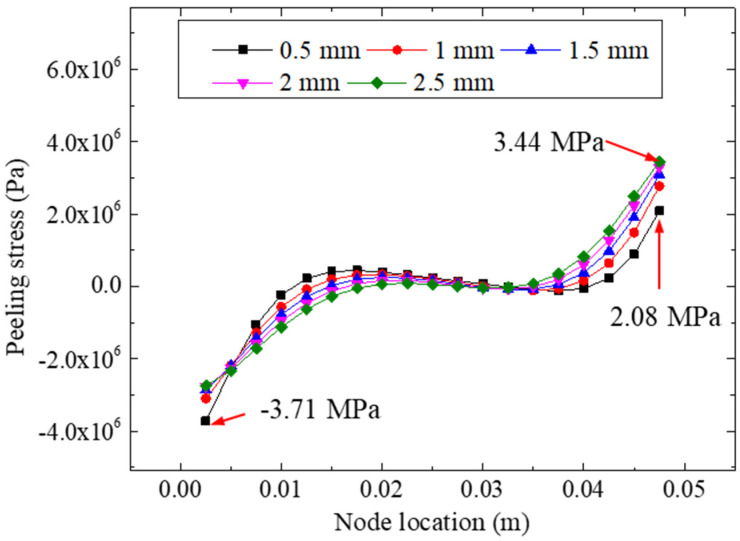
Peeling stresses for different bonding thicknesses.

**Figure 16 materials-15-07787-f016:**
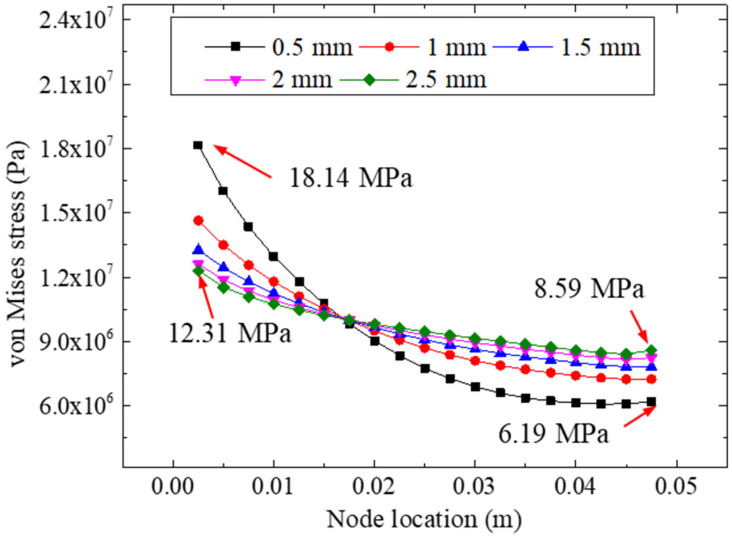
The von Mises stresses for different bonding thicknesses.

**Figure 17 materials-15-07787-f017:**
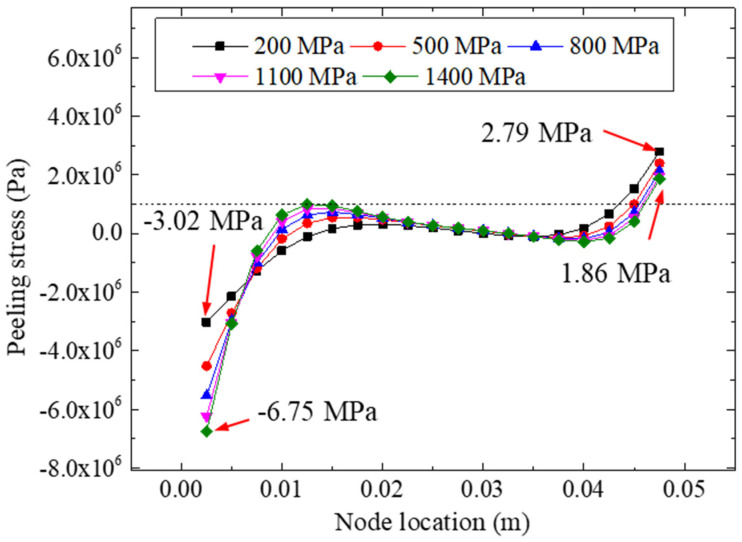
Peeling stresses for different adhesive modulus.

**Figure 18 materials-15-07787-f018:**
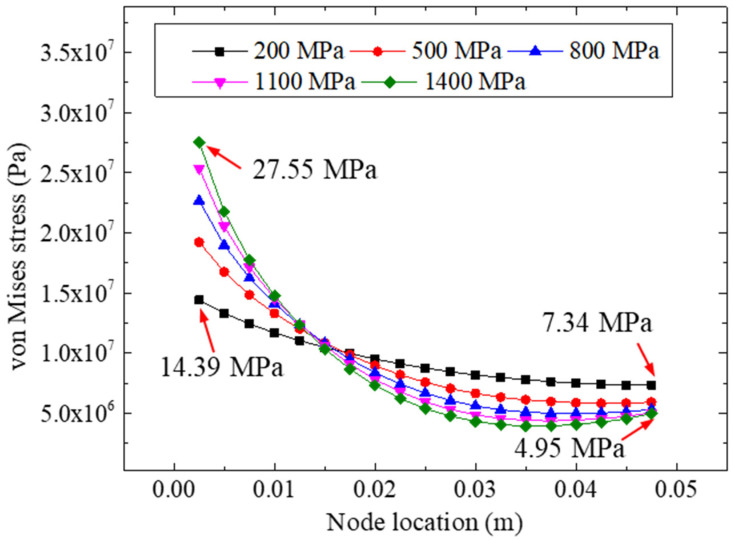
The von Mises stresses for different adhesive modulus.

**Table 1 materials-15-07787-t001:** Mechanical properties of the GFRP, adhesive, and steel.

Mechanical Parameter	GFRP	Adhesive *	Steel
Young’s modulus, Mpa	15,400 (longitudinal direction) 6850 (transverse direction) 7630 (thickness direction)	213.24 * (Secant Young’s modulus)	204,000
Strength, Mpa	291.1 (longitudinal direction) 125.3 (transverse direction) 144.6 (thickness direction)	18.85 *	291.3 (Yield stress)
Poisson’s ratio	0.37	0.40 (According to product manual)	0.3

* I mechanical properties were determined by a tensile test.

**Table 2 materials-15-07787-t002:** Stress analysis results for five elastic moduli.

Elastic Modulus of FRP Plates/Gpa	Ultimate Peeling Stresses for Path II/Mpa	Von Mises Stresses for Path II/Mpa
10	−3.80	16.96 (left)/6.15 (right)
50	−1.95	11.02 (left)/9.25 (right)
100	−1.61	10.09 (left)/9.84 (right)
200	−1.41	9.59 (left)/10.18 (right)
500	−1.28	9.29 (left)/10.41 (right)

## Data Availability

Not applicable.

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
