# Peer review of "Parameter Study of Interfacial Capacities for FRP–Steel Bonded Joints Based on 3D FE Modeling"

_materials, 2022, doi:10.3390/ma15217787_

Round 1

Reviewer 1 Report

1.      The abstract requires the addition of quantitative results.

2.      Please end your abstract with a "take-home" message.

3.      Keywords should be reordered based on alphabetical order.

4.      Make the each of keywords with lowercase font following MDPI format, revise it.

5.      Please do not use abbreviations in keywords.

6.      What is the novel of the present study? It works have been widely studied in the past. Nothing something really new in the present form. The lack of novel seems to make the present study like to replication/modified study. The authors need to detail their novelty in the introduction section. It is a major concern for rejecting this paper.

7.      In order to highlight the gaps in the literature that the most recent research aims to fill, it is crucial to review the benefits, novelty, and limitations of earlier studies in the introduction.

8.      “Von Mises stress” is should be “von Mises stress” with lowercase on “v” from “von”, correct it.

9.      The authors need to explain the advantages of investigation of computational simulation via finite element analysis compared with experimental texting such as faster results and lower cost. Authors must address this crucial aspect in the introduction and/or discussion section. In addition, to support this explanation, the recommended literature should be included as follows: Ammarullah, M. I.; Santoso, G.; Sugiharto, S.; Supriyono, T.; Kurdi, O.; Tauviqirrahman, M.; Winarni, T. I.; Jamari, J. Tresca Stress Study of CoCrMo-on-CoCrMo Bearings Based on Body Mass Index Using 2D Computational Model. Jurnal Tribologi 2022, 33, 31–8. https://jurnaltribologi.mytribos.org/v33/JT-33-31-38.pdf

10.   Line 108, what a version of ANSYS? It should be mention including country and corporation.

Author Response

Comment #1. The abstract requires the addition of quantitative results.

Response: To address this comment, the following sentences are given in the revised manuscript.

“This paper investigated the stress distribution of an adhesive layer for GFRP-steel bonded joints under 22.48 kN tensile loading by 3-dimensional numerical simulation. Firstly, the stress analysis of three paths was conducted, and by comparison, path II (through the middle layer of the bonding layer) was adopted as the analyzing path. Furthermore, a systemically parametric study was conducted and discussed. For the joints with different FRP elastic modulus, the minimum value of normal peeling stress was calculated as -3.80MPa by the FRP for 10 GPa, showing a significantly severe stress concentration of FRP for 10 GPa. Analysis of von Mises stress proved that the increase of FRP stiffness can reduce the stress concentration of the adhesive layer effectively. The study of the effect of bonding lengths indicated that more uniform peeling stress distribution can be resulted from the longest bonding size, and the biggest peeling stress of 6.54 MPa was calculated by the bonding length of 30 mm. Further parameter analysis showed that the stress concentration of the adhesive layer could be influenced by FRP thickness, bonding thickness, and elastic modulus of the adhesive layer.” (Lines:1~15)

Comment #2. Please end your abstract with a "take-home" message.

Response: To address this comment, the following sentences are given in the revised manuscript.

“This paper investigated the stress distribution of an adhesive layer for GFRP-steel bonded joints under 22.48 kN tensile loading by 3-dimensional numerical simulation. Firstly, the stress analysis of three paths was conducted, and by comparison, path II (through the middle layer of the bonding layer) was adopted as the analyzing path. Furthermore, a systemically parametric study was conducted and discussed. For the joints with different FRP elastic modulus, the minimum value of normal peeling stress was calculated as -3.80MPa by the FRP for 10 GPa, showing a significantly severe stress concentration of FRP for 10 GPa. Analysis of von Mises stress proved that the increase of FRP stiffness can reduce the stress concentration of the adhesive layer effectively. The study of the effect of bonding lengths indicated that more uniform peeling stress distribution can be resulted from the longest bonding size, and the biggest peeling stress of 6.54 MPa was calculated by the bonding length of 30 mm. Further parameter analysis showed that the stress concentration of the adhesive layer could be influenced by FRP thickness, bonding thickness, and elastic modulus of the adhesive layer.” (Lines:1~15)

Comment #3. Keywords should be reordered based on alphabetical order.

Response: To address this comment, the Keywords have been revised as following:

“adhesive; equivalent stress; fiber reinforced polymer-steel; finite element model; peeling stress” (Lines:16~17)

Comment #4. Make the each of keywords with lowercase font following MDPI format, revise it.

Response: To address this comment, the Keywords have been revised as following:

“adhesive; equivalent stress; fiber reinforced polymer-steel; finite element model; peeling stress” (Lines:16~17)

Comment #5. Please do not use abbreviations in keywords.

Response: To address this comment, the Keywords have been revised as following:

“adhesive; equivalent stress; fiber reinforced polymer-steel; finite element model; peeling stress” (Lines:16~17)

Comment #6. What is the novel of the present study? It works have been widely studied in the past. Nothing something really new in the present form. The lack of novel seems to make the present study like to replication/modified study. The authors need to detail their novelty in the introduction section. It is a major concern for rejecting this paper.

Response: To address this comment, the following sentences are given in the revised manuscript.

“The above researchers have studied the performance change law of FRP-steel bonded structure based on macro performance analysis methods. However, considering the complexity and difficulty in quantitative detection of the research on the interface characteristics of the bonding structure (such as the stress distribution inside the bonding layer, the internal trend of failure, etc.) [17-21], the current research on the failure characteristics and change trend of micro details of the bonding layer is still relatively limited. To fill this gap, this paper presented the studies of interfacial stress for GFRP-steel double-strap bonded joints via the finite element (FE) method. Based on this study, the peeling stress and von Mises stress of the adhesive layer under different bonding parameters were analyzed, by which the failure mechanism of the steel-FRP bonding interface was discussed and expounded rationally.” (Lines:47~55)

Comment #7. In order to highlight the gaps in the literature that the most recent research aims to fill, it is crucial to review the benefits, novelty, and limitations of earlier studies in the introduction.

Response: To address this comment, the following sentences are given in the revised manuscript.

“The above researchers have studied the performance change law of FRP-steel bonded structure based on macro performance analysis methods. However, considering the complexity and difficulty in quantitative detection of the research on the interface characteristics of the bonding structure (such as the stress distribution inside the bonding layer, the internal trend of failure, etc.) [17-21], the current research on the failure characteristics and change trend of micro details of the bonding layer is still relatively limited. To fill this gap, this paper presented the studies of interfacial stress for GFRP-steel double-strap bonded joints via the finite element (FE) method. Based on this study, the peeling stress and von Mises stress of the adhesive layer under different bonding parameters were analyzed, by which the failure mechanism of the steel-FRP bonding interface was discussed and expounded rationally.” (Lines:47~55)

Comment #8. Von Mises stress” is should be “von Mises stress” with lowercase on “v” from “von”, correct it.

Response: To address this comment, the “Von Mises stress” have been changed as “von Mises stress” in the revised manuscript.

Comment #9. The authors need to explain the advantages of investigation of computational simulation via finite element analysis compared with experimental texting such as faster results and lower cost. Authors must address this crucial aspect in the introduction and/or discussion section. In addition, to support this explanation, the recommended literature should be included as follows: Ammarullah, M. I.; Santoso, G.; Sugiharto, S.; Supriyono, T.; Kurdi, O.; Tauviqirrahman, M.; Winarni, T. I.; Jamari, J. Tresca Stress Study of CoCrMo-on-CoCrMo Bearings Based on Body Mass Index Using 2D Computational Model. Jurnal Tribologi 2022, 33, 31–8. https://jurnaltribologi.mytribos.org/v33/JT-33-31-38.pdf

Response: to address this comment, the following sentences have been added in the revised paper.

“Although the experimental research can obtain more intuitive and reliable conclusions, it is difficult to monitor the internal mechanical characteristics and change mechanism of the component. Benefited from its convenience and economy, the finite element analysis method makes up for the shortage of the experimental research method [23-28], so it is widely used in the engineering research field.” (Lines:94~97)

Comment #10. Line 108, what a version of ANSYS? It should be mention including country and corporation.

Response: To address this comment, the sentences have been revised as following.

“To analyze the interfacial stress distribution of the GFRP-steel joints with different bonding parameters, 3-dimension finite element models were developed by using the FE software package ANSYS 14.5, as shown in Fig. 3. (Lines: 98~100)

Reviewer 2 Report

The manuscript may be accepted if the following errors are corrected:

1. How many samples are required to get the results shown in Figure 2, the manuscript needs further discussion on this

2. Other results in the manuscript with experimental use should also state the number of samples tested.

3. Symbol "node location" on the Ox axis of drawings 9.10: the manuscript needs to clearly describe from which position to which position, and why not draw from the position with coordinates 0 (mm)

4. The manuscript should cite some of the following articles related to multilayer structures

- DOI: 10.1007/978-981-10-7149-2_3

- https://doi.org/10.1155/2019/6814367

- https://doi.org/10.3390/ma12040598

Author Response

Comment #1 How many samples are required to get the results shown in Figure 2, the manuscript needs further discussion on this.

Response: Thank you for the comment, one adhesive sample was tested and the capacity of 18.85 MPa (tensile strength) was presented in the following paper:

Liu, J.; Guo, T.; Feng, D.M.; Liu, Z.X. Fatigue performance of rib-to-deck joints strengthened with FRP angles. J. Bridge. Eng. 2018, 23(9), 04018060.

Comment #2: Other results in the manuscript with experimental use should also state the number of samples tested.

Response: Thank you for the comment, one sample of the FRP-steel bonded specimen was tested and the capacity of 22.48 kN (tensile strength) was presented in the following paper:

Liu, J.; Guo, T.; Hebdon, M.H.; Liu, Z.X.; Wang, L.B. Bonding Behaviors of GFRP/Steel Bonded Joints after Wet–Dry Cyclic and Hygrothermal Curing. Appl. Sci-basel. 2020, 10(16),5411.

Comment #3. Symbol "node location" on the Ox axis of drawings 9.10: the manuscript needs to clearly describe from which position to which position, and why not draw from the position with coordinates 0 (mm)

Response: Thank you for your comment. The position of the extracted node starts from 0.0025 m (on the Ox axis), because the node stress gets the extreme value at this position.

Comment #4. The manuscript should cite some of the following articles related to multilayer structures

- DOI: 10.1007/978-981-10-7149-2_3

- https://doi.org/10.1155/2019/6814367

- https://doi.org/10.3390/ma12040598

Response: To address this comment, these articles have been added in the revised manuscript.

Duc, N. D., Trinh, T. D., Do, T. V., & Doan, D. H. On the buckling behavior of multi-cracked FGM plates. In International Conference on Advances in Computational Mechanics, 2017, 29-45. Springer, Singapore.

“Nam, V. H., Nam, N. H., Vinh, P. V., Khoa, D. N., Thom, D. V., & Minh, P. V. A new efficient modified first-order shear model for static bending and vibration behaviors of two-layer composite plate. Adv. Civ. Eng., 2019:6814367”

“Nguyen, H. N., Nguyen, T. Y., Tran, K. V., Tran, T. T., Nguyen, T. T., Phan, V. D., & Do, T. V. A finite element model for dynamic analysis of triple-layer composite plates with layers connected by shear connectors subjected to moving load. Materials, 2019, 12(4), 598.”

Reviewer 3 Report

Interfacial capabilities of bonded joints between FRP and steel have been studied in the literature. To improve the Introduction, I suggest to cite also the following work:

Yang et al. 2021 - http://dx.doi.org/10.1016/j.tws.2012.07.020

Which and clearly highlight the novelty of your work compared to these ones.

The work presents only a numerical investigation without any experimental validation. To improve the reliability of the FE model, it is important to make a calibration through convergence of the mesh.

A simple but accurate and efficient numerical model for the analysis of bonded joints is presented in doi:10.1016/j.ijadhadh.2012.01.012 and DOI:10.1163/156856109X433027. This can be a reference for the improvement of your FE model.

A large number of diagrams are presented: I suggest to remove some of them if possible.

Author Response

Comment #1. Interfacial capabilities of bonded joints between FRP and steel have been studied in the literature. To improve the Introduction, I suggest to cite also the following work:

Yang et al. 2021 - http://dx.doi.org/10.1016/j.tws.2012.07.020

Response: To address this comment, the article has been added in the revised paper manuscript.

“Yang, J. Q., Smith, S. T., & Feng, P. (2013). Effect of FRP-to-steel bonded joint configuration on interfacial stresses: Finite element investigation. Thin Wall. Struct., 2012, 62, 215-228.”

Comment #2. Which and clearly highlight the novelty of your work compared to these ones.

Response: To address this comment, the following sentences are given in the revised manuscript.

“The above researchers have studied the performance change law of FRP-steel bonded structure based on macro performance analysis methods. However, considering the complexity and difficulty in quantitative detection of the research on the interface characteristics of the bonding structure (such as the stress distribution inside the bonding layer, the internal trend of failure, etc.) [17-21], the current research on the failure characteristics and change trend of micro details of the bonding layer is still relatively limited. To fill this gap, this paper presented the studies of interfacial stress for GFRP-steel double-strap bonded joints via the finite element (FE) method. Based on this study, the peeling stress and von Mises stress of the adhesive layer under different bonding parameters were analyzed, by which the failure mechanism of the steel-FRP bonding interface was discussed and expounded rationally.” (Lines:47~55)

Comment #3. The work presents only a numerical investigation without any experimental validation. To improve the reliability of the FE model, it is important to make a calibration through convergence of the mesh.

Response #3: Thank you for your comment, and the numerical analysis results in this paper have been compared with the previous experimental results (the stress value of the bonding layer under the failure load of the bonding joint).

Comment #4. A simple but accurate and efficient numerical model for the analysis of bonded joints is presented in doi:10.1016/j.ijadhadh.2012.01.012 and DOI:10.1163/156856109X433027. This can be a reference for the improvement of your FE model.

Response: To address this comment, the articles have been added in the revised paper manuscript.

“Castagnetti, D., Spaggiari, A., & Dragoni, E. Assessment of the Cohesive Contact method for the analysis of thin-walled bonded structures. Int. J. adhes. adhes., 2012, 37, 112-120.”

“Castagnetti, D., Dragoni, E., & Spaggiari, A. Efficient post-elastic analysis of bonded joints by standard finite element techniques. J. adhes. Sci. technol., 2009,23(10-11), 1459-1476.”

Comment #5. A large number of diagrams are presented: I suggest to remove some of them if possible.

Response: Thank you for your comment, and the redundant figures have been deleted.

Round 2

Reviewer 1 Report

Well done by the authors, I have some other comments that need to be addressed as follows:

1.      Explain in detail about meshing strategies for finite element dissertation.

2.      What type of load? Static of dynamic? Explain specifically.

3.      It is unclear it is implicit or explicit simulation conducted by the authors.

4.      Materials assumption needs more explanation, it is homogeneous/heterogeneous, linear/nonlinear plastic/elastic, and other. Explain also the importance of materials assumption in computational simulation, additional reference from MDPI is needed as follows: Ammarullah, M. I.; Santoso, G.; Sugiharto, S.; Supriyono, T.; Wibowo, D. B.; Kurdi, O.; Tauviqirrahman, M.; Jamari, J. Minimizing Risk of Failure from Ceramic-on-Ceramic Total Hip Prosthesis by Selecting Ceramic Materials Based on Tresca Stress. Sustainability 2022, 14, 13413. https://doi.org/10.3390/su142013413

5.      To let the reader, comprehend the workflow of the current study, the authors could include extra illustrations as a type of figure in the materials and methods rather than simply the main text as a present form.

6.      Findings must be compared to similar past research.

7.      What is the limitation of the present work? Please include it before the conclusion section.

8.      Add more detail to the conclusion by structuring it as a paragraph rather than in point-by-point as a present form.

9.      In the conclusion section, further research must be discussed.

10.   The reference should be enriched with literature from the last five years. Literature published by MDPI is strongly recommended.

11.   Throughout the whole manuscript, the authors sometimes wrote paragraphs with just one or two phrases, which made the explanation difficult to understand. To make their explanation a full paragraph, the writers should expand it. It is advised to use at least three sentences in a paragraph, with the primary sentence coming first and the supporting sentences coming after.

12.   Due to grammatical and language issues, the authors need to proofread the present work.

13.   Please ensure that the authors followed the MDPI format correctly; modify the current form and recheck, as well as any other problems that have been highlighted.

14.   After revision, provide a graphical abstract for submission.

15.   Provide a graphical abstract after revision in resubmit.

Author Response

Comment #1:Well done by the authors, I have some other comments that need to be addressed as follows: Explain in detail about meshing strategies for finite element dissertation.

Response: To address this comment, the following sentences are given in the manuscript.

“The GFRP plates, adhesive layers, and steel plates were modeled using a 3D 8-node solid element (i.e., Solid 185 in ANSYS), having three degrees of freedom (DOFs) at each node (translations in the x, y, and z), and meshed with a size of 0.5 × 0.5 × 0.5 mm. Under this meshing strategy, the bonding layer is simulated by a two-layer hexahedral element” (Lines:119~123)

Comment #2. What type of load? Static of dynamic? Explain specifically.

Response: Thank you for your comment, the following sentences for the type of load are given in the manuscript.

“During the static tensile testing, the load was applied through the two ends of the specimen, with the displacement-control mode set at a rate of 1 mm/min, and a tensile capacity of 22.48 kN was adopted [22](Lines:103~105)

Comment #3. It is unclear it is implicit or explicit simulation conducted by the authors.

Response: To address this comment, the sentence has been added as following:

“For all finite element analyses, an implicit calculation method is performed.” (Lines:117~118)

Comment #4. Materials assumption needs more explanation, it is homogeneous/heterogeneous, linear/nonlinear plastic/elastic, and other. Explain also the importance of materials assumption in computational simulation, additional reference from MDPI is needed as follows: Ammarullah, M. I.; Santoso, G.; Sugiharto, S.; Supriyono, T.; Wibowo, D. B.; Kurdi, O.; Tauviqirrahman, M.; Jamari, J. Minimizing Risk of Failure from Ceramic-on-Ceramic Total Hip Prosthesis by Selecting Ceramic Materials Based on Tresca Stress. Sustainability 2022, 14, 13413. https://doi.org/10.3390/su142013413

Response: To address this comment, the sentences have been added in the revised manuscript.

“The steel, adhesive, and GFRPs were assumed to be homogeneous and linearly elastic, which were defined according to the values in Table 1.” (Lines:124~125)

Ammarullah, M.I., Santoso, G., Sugiharto, S., Supriyono, T., Wibowo, D.B., Kurdi, O., Tauviqirrahman, M. and Jamari, J., Minimizing Risk of Failure from Ceramic-on-Ceramic Total Hip Prosthesis by Selecting Ceramic Materials Based on Tresca Stress. Sustainability, 2022.14(20), 13413. (Lines:470~471)

Comment #5. To let the reader, comprehend the workflow of the current study, the authors could include extra illustrations as a type of figure in the materials and methods rather than simply the main text as a present form.

Response: To address this comment, the workflow has been added, please see the Figure 8 in the revised manuscript.

Comment #6. Findings must be compared to similar past research.

Response: To address this comment, the following comparison is given in the revised manuscript.

The changes of the extreme normal peeling stress show that the stress concentration at the end of the bonding layer decreases with the increase of the FRP plate stiffness and the bonding strength of the joint could be improved accordingly [36]. (Lines:217~219)

“The homogenization of peeling stress can effectively improve the strength of speci-mens, and similar conclusions have been studied and reported [37].” (Lines:282~283)

Comment #7. What is the limitation of the present work? Please include it before the conclusion section.

Response: To address this comment, the following sentences are given in the revised manuscript.

“It is worth noting that, this study is aimed at the bonding stresses of FRP-steel double-strap bonded joints based on linear elastic assumption and cohesive failure mode, therefore, the failure mechanism of FRP-steel composite members made of nonlinear materials with other failure modes (except for cohesive failure) needs further research.” (Lines:406~410)

Comment #8. Add more detail to the conclusion by structuring it as a paragraph rather than in point-by-point as a present form.

Response: Thank you for your comment, the conclusion has been checked and improved as following:

“The normal peeling stress distribution of the bonding layer is more uniform for the specimens bonded with FRP materials with higher elastic modulus.” (Lines:213~214)

“By comparing Fig. 10 and Fig. 12, similar trends of stress curves could be found, for example, the stress distribution of specimens made of the more rigid FRP plates (e.g., 7-mm FRP and 500 GPa) is more uniform than that of the more flexible FRP plates (e.g., 3-mm FRP and 10 GPa).” (Lines:262~265)

“In addition, for the specimens with a bonding length greater than 90 mm, the increase of bonding length has not so remarkable effect on the bonding stresses, for example, the extreme peeling stress of the 150-mm bonding specimen (0.12 MPa) was only 0.60 MPa less than that for 90-mm bonding one (0.72 MPa).” (Lines:285~288)

“By analyzing the stress values of different bonding thicknesses, it is found that the change of the bonding thickness can adjust the peeling stress of the interface to a certain extent, which is specifically reflected in: the thinner the bonding layer is, the extreme value of peeling stress on the left side (near the centerline) is more prominent, and conversely, the thicker the bonding layer is, the extreme value on the right side is more prominent.” (Lines:318~323)

“However, for the left-side, the minimum peeling stresses were -3.02 (200 MPa), -4.53 (500 MPa), -5.53 (800 MPa), -6.24 (1100 MPa), and -6.75 (1400 MPa) MPa respectively, indicating that the peeling stress on the left-side of the bonding layer is more sensitive to the stiffness of the adhesives.” (Lines:356~359)

Comment #9. In the conclusion section, further research must be discussed.

Response: to address this comment, the following sentence has been added in the revised paper.

“It is worth noting that, this study is aimed at the bonding stresses of FRP-steel double-strap bonded joints based on linear elastic assumption and cohesive failure mode, therefore, the failure mechanism of FRP-steel composite members made of nonlinear materials with other failure modes (except for cohesive failure) needs further research.” (Lines:406~410)

Comment #10. The reference should be enriched with literature from the last five years. Literature published by MDPI is strongly recommended.

Response: To address this comment, the literatures have been added as following.

Nguyen, H. N., Nguyen, T. Y., Tran, K. V., Tran, T. T., Nguyen, T. T., Phan, V. D., & Do, T. V. A finite element model for dynamic analysis of triple-layer composite plates with layers connected by shear connectors subjected to moving load. Materials, 2019, 12(4), 598. (Lines: 449~451)

Liu, J.; Guo, T.; Hebdon, M.H.; Liu, Z.X.; Wang, L.B. Bonding Behaviors of GFRP/Steel Bonded Joints after Wet–Dry Cyclic and Hygrothermal Curing. Appl. Sci-basel. 2020, 10(16),5411. (Lines: 456~457)

Wang, J.; Dai, Q.; Lautala, P.; Yao, H.; Si, R. Rail Sample Laboratory Evaluation of Eddy Current Rail Inspection Sustainable System. Sustainability, 2022, 14, 11568. (Lines: 460~461)

Chen, S., Wang, J., Li, Q., Zhang, W., & Yan, C. The Investigation of Volatile Organic Compounds (VOCs) Emissions in Environmentally Friendly Modified Asphalt. Polymers, 2022, 14(17), 3459. (Lines: 462~463)

Ammarullah, M.I., Santoso, G., Sugiharto, S., Supriyono, T., Wibowo, D.B., Kurdi, O., Tauviqirrahman, M. and Jamari, J., Minimizing Risk of Failure from Ceramic-on-Ceramic Total Hip Prosthesis by Selecting Ceramic Materials Based on Tresca Stress. Sustainability, 2022.14(20), 13413. (Lines: 470~472)

Comment #11. Throughout the whole manuscript, the authors sometimes wrote paragraphs with just one or two phrases, which made the explanation difficult to understand. To make their explanation a full paragraph, the writers should expand it. It is advised to use at least three sentences in a paragraph, with the primary sentence coming first and the supporting sentences coming after.

Response: Thank you for the comment, the paragraphs have been improved.

“According to Fig. 9, it is found that the elastic modulus of FRP can significantly affect the normal peeling stress distribution of the bonding layer. The normal peeling stress distribution of the bonding layer is more uniform for the specimens bonded with FRP materials with higher elastic modulus.” (Lines:211~213)

“With the increase of FRP stiffness, the von Mises stresses on the left-side of the adhesive layer decrease gradually, while the stresses on the right-side increase gradually.” (Lines:227~229)

“From Fig. 11, it is observed that the peeling stress could be influenced by FRP thickness. As the bonding length increases, the extreme peeling stress of the right-side increases accordingly.” (Lines:247~249)

“It is observed that the FRP thickness may affect the extreme equivalent stress of the adhesive layer.” (Lines:258~259)

“It is found that the bonding length may influence the peeling stress of the adhesive layer significantly. The biggest peeling stress was calculated by the bonding length of 30 mm, with the extreme stress of 6.54 MPa on its right-side. As the bonding length increases, the extreme peeling stress decreases accordingly.” (Lines:276~279)

“It can be seen from the curves that, for each specimen with a specific bonding thickness, with the increase of the node coordinates, the equivalent stress at the interface shows a trend of decrease, showing that under the current specimen parameters (i.e., bonding length and stiffness), the early failure of the left bonding zone is the main cause of the failure of the composite structures.” (Lines:327~331)

“It is found that the elastic modulus of the adhesive layer has a significant influence on the peeling stress of the bonded specimen. For different adhesive moduli (i.e., 200, 500, 800, 1 100, and 1 400 MPa), the maximum peeling stresses of the right-side were 2.79, 2.38, 2.12, 1.96, and 1.86 MPa respectively, showing that the elastic modulus of adhe-sive layer has little influence on the free end (i.e., the right-side of the bonded speci-mens).” (Lines:351~356)

Comment #12. Due to grammatical and language issues, the authors need to proofread the present work.

Response: Thank you for the comment, the language has been improved.

Comment #13. Please ensure that the authors followed the MDPI format correctly; modify the current form and recheck, as well as any other problems that have been highlighted.

Response: Thank you for the comment, the format has been checked carefully.

Comment #14. After revision, provide a graphical abstract for submission.

Response: Thank you for the comment, the abstract has been improved carefully.

“This paper investigated the stress distribution of an adhesive layer for GFRP-steel bonded joints under 22.48 kN tensile loading by 3-dimensional numerical simulation. Firstly, the stress analysis of three paths was conducted, and by comparison, path II (through the middle layer of the bonding layer) was adopted as the analyzing path. Furthermore, a systemically parametric study of the effects of FRP stiffness (i.e., elastic modulus and thickness), bonding length, adhesive thickness, and adhesive modulus was conducted. For the joints with different FRP elastic modulus, the minimum value of normal peeling stress was calculated as -3.80MPa by the FRP for 10 GPa, showing a significantly severe stress concentration of FRP for 10 GPa. Analysis of von Mises stress proved that the increase of FRP stiffness can reduce the stress concentration of the adhesive layer effectively. The study of the effect of bonding lengths indicated that more uniform peeling stress distribution can be resulted from the longest bonding size, and the biggest peeling stress of 6.54 MPa was calculated by the bonding length of 30 mm. Further parameter analysis showed that the stress concentration of the adhesive layer could be influenced by FRP thickness, bonding thickness, and elastic modulus of the adhesive layer.” (Lines: 10~23)

Comment #15. Provide a graphical abstract after revision in resubmit.

Response: Thank you for the comment, the abstract has been improved carefully.

“This paper investigated the stress distribution of an adhesive layer for GFRP-steel bonded joints under 22.48 kN tensile loading by 3-dimensional numerical simulation. Firstly, the stress analysis of three paths was conducted, and by comparison, path II (through the middle layer of the bonding layer) was adopted as the analyzing path. Furthermore, a systemically parametric study of the effects of FRP stiffness (i.e., elastic modulus and thickness), bonding length, adhesive thickness, and adhesive modulus was conducted. For the joints with different FRP elastic modulus, the minimum value of normal peeling stress was calculated as -3.80MPa by the FRP for 10 GPa, showing a significant severe stress concentration of FRP for 10 GPa. Analysis of von Mises stress proved that the increase of FRP stiffness can reduce the stress concentration of the adhesive layer effectively. The study of the effect of bonding lengths indicated that more uniform peeling stress distribution can be resulted from the longest bonding size, and the biggest peeling stress of 6.54 MPa was calculated by the bonding length of 30 mm. Further parameter analysis showed that the stress concentration of the adhesive layer could be influenced by FRP thickness, bonding thickness, and elastic modulus of the adhesive layer.” (Lines: 10~23)

Reviewer 3 Report

The authors edited the paper according to the reviewers' suggestions

Author Response

Thank you for your comment, the manuscript has been checked and improved according to the reviewers' suggestions.

Round 3

Reviewer 1 Report

Highly appreciate to The Authors for their well-done revision.